# An Alzheimer’s Disease Patient-Derived Olfactory Stem Cell Model Identifies Gene Expression Changes Associated with Cognition

**DOI:** 10.3390/cells11203258

**Published:** 2022-10-17

**Authors:** Laura M. Rantanen, Maina Bitar, Riikka Lampinen, Romal Stewart, Hazel Quek, Lotta E. Oikari, Carla Cunί-Lόpez, Ratneswary Sutharsan, Gayathri Thillaiyampalam, Jamila Iqbal, Daniel Russell, Elina Penttilä, Heikki Löppönen, Juha-Matti Lehtola, Toni Saari, Sanna Hannonen, Anne M. Koivisto, Larisa M. Haupt, Alan Mackay-Sim, Alexandre S. Cristino, Katja M. Kanninen, Anthony R. White

**Affiliations:** 1Mental Health and Neuroscience, QIMR Berghofer Medical Research Institute, Brisbane, QLD 4006, Australia; 2Genomics Research Centre, Centre for Genomics and Personalised Health, School of Biomedical Sciences, Queensland University of Technology (QUT), Brisbane, QLD 4059, Australia; 3A.I. Virtanen Institute for Molecular Sciences, University of Eastern Finland, 70210 Kuopio, Finland; 4Griffith Institute for Drug Discovery, Griffith University, Brisbane, QLD 4111, Australia; 5Department of Otorhinolaryngology, University of Eastern Finland, Kuopio University Hospital, 70210 Kuopio, Finland; 6Brain Research Unit, Department of Neurology, School of Medicine, University of Eastern Finland, 70210 Kuopio, Finland; 7Department of Neurology, Neuro Centre, Kuopio University Hospital, 70210 Kuopio, Finland; 8Department of Neurology and Geriatrics, Helsinki University Hospital and Neurosciences, Faculty of Medicine, University of Helsinki, 00014 Helsinki, Finland; 9The University of Queensland Diamantina Institute, Brisbane, QLD 4102, Australia; 10Faculty of Medicine, University of Queensland, Brisbane, QLD 4072, Australia

**Keywords:** Alzheimer’s disease, mild cognitive impairment, patient-derived olfactory mucosa, olfactory neurosphere-derived cells, RNA Sequencing, *AKAP6*, cognition, aging

## Abstract

An early symptom of Alzheimer’s disease (AD) is an impaired sense of smell, for which the molecular basis remains elusive. Here, we generated human olfactory neurosphere-derived (ONS) cells from people with AD and mild cognitive impairment (MCI), and performed global RNA sequencing to determine gene expression changes. ONS cells expressed markers of neuroglial differentiation, providing a unique cellular model to explore changes of early AD-associated pathways. Our transcriptomics data from ONS cells revealed differentially expressed genes (DEGs) associated with cognitive processes in AD cells compared to MCI, or matched healthy controls (HC). A-Kinase Anchoring Protein 6 (*AKAP6*) was the most significantly altered gene in AD compared to both MCI and HC, and has been linked to cognitive function. The greatest change in gene expression of all DEGs occurred between AD and MCI. Gene pathway analysis revealed defects in multiple cellular processes with aging, intellectual deficiency and alternative splicing being the most significantly dysregulated in AD ONS cells. Our results demonstrate that ONS cells can provide a cellular model for AD that recapitulates disease-associated differences. We have revealed potential novel genes, including *AKAP6* that may have a role in AD, particularly MCI to AD transition, and should be further examined.

## 1. Introduction

Alzheimer’s disease (AD) affects over 55 million people worldwide. There are currently no disease-modifying treatments, aside from the recently approved anti-amyloid antibody, Aduhelm^TM^. While mutations in several genes cause a small number of AD cases [1,2], the majority of cases (known as late-onset AD, LOAD) are multifactorial, involving risk genes, epigenetic changes, and lifestyle and environmental factors that are yet to be fully elucidated [3]. The most significant risk factors for LOAD include aging and expression of allele ε4 of the apolipoprotein E (*APOE4*) genotype [4]. The primary hallmark pathologies of the AD brain are progressive extracellular accumulation of amyloid-β (Aβ) plaques, and intracellular neurofibrillary tangles (NFTs) [5]. The olfactory system is responsible for our sense of smell (olfaction) and is integrated into the brain via the olfactory nerve with olfactory input known to affect cognitive function [6,7]. Olfaction is either partially (hyposmia) or completely (anosmia) impaired in AD, and may be associated with early stages in disease pathology. Despite early changes in olfaction, together with the appearance of amyloid plaques, NFTs and oxidative stress in olfactory cells, the role of the olfactory system in AD remains poorly understood [8,9].

The olfactory system consists of the olfactory mucosa (OM), located in the upper region of the nasal cavity, where the most superficial layer (olfactory epithelium, OE) is in contact with the external environment. The OE is continuously exposed to potentially harmful agents, including allergens and air pollution, which may contribute to accumulating damage and epigenetic changes to olfactory DNA, potentially with the ability to modify the risk for AD occurrence [10,11]. The OE contains stem cells, including horizontal and globose basal cells which can be readily accessed by routine biopsy, and grown in vitro to form olfactory neurospheres and human neurosphere-derived (ONS) cells [12]. ONS cells can differentiate into neuron- and glial-like cells, and cells of non-ectodermal lineage [13,14,15] without genetic reprogramming. The ONS cell model has been used to generate robust disease-specific phenotypes relevant to several neurodegenerative disorders [8,9,14,16,17,18,19,20], and it provides a unique opportunity to investigate transcriptomic profiling of AD olfactory tissue upon establishment of a neuro-glial phenotype. The unique brain-like microenvironment provided by the ONS model has the potential to identify gene expression changes associated with AD, rather than simply reflecting the original olfactory epithelial gene expression.

In this study, we generated ONS cell cultures using olfactory mucosal biopsies from age-matched cognitively healthy individuals (HC), patients with LOAD (AD), and individuals with mild cognitive impairment (MCI), a prodromal AD condition. We examined the growth characteristics of the olfactory neurospheres and whole transcriptome of ONS cells to determine gene expression changes in AD and MCI individuals.

## 2. Materials and Methods

### 2.1. Ethical Considerations

Olfactory biopsies were performed with approval of the Human Research Ethics Committees (HRECs), of Northern Savo Hospital District (permit number 536/2017), and QIMR Berghofer Medical Research Institute (P2197). Written informed consent was collected from all subjects and proxy consent from family members of persons with mild AD dementia. All research adhered to ethical guidelines on human research outlined by the National Health and Medical Research Council of Australia (NHMRC), QIMR Berghofer Medical Research Institute, and Northern Savo Hospital District.

### 2.2. Patients and Nasal Biopsy

Volunteers clinically diagnosed with MCI (*n* = 6) and AD mild dementia (CDR 1) (*n* = 6) were recruited together with cognitively healthy control subjects (*n* = 6) via the Brain Research Unit, Department of Neurology, University of Eastern Finland. Prior to study recruitment, diagnostic examinations had been carried out at the Brain Research Unit or at the Department of Neurology, Kuopio University Hospital as previously described [21,22]. Their biopsy samples were collected and processed as previously described [21]. Briefly, a piece of OM was biopsied from the nasal septum and kept on ice in Dulbecco’s Modified Eagle Medium/Ham F-12 (DMEM/F12) (#11320033)-based growth medium containing 10% heat-inactivated fetal bovine serum (FBS) (#10500064) and 1% Penicillin-Streptomycin solution (P/S) (#15140122) until processed in the laboratory (All reagents Thermo Fisher Scientific, Waltham, MA, USA). Primary OM cell cultures were established according to the published protocol [21,23]. The age of the patients with AD and cognitively healthy control subjects age averaged ~71 years and 70 years, respectively, while the MCI group averaged 72 years. For the AD and control group there were four females and two male study subjects, whereas in the MCI group, there were two females and four males. Based on blood APOE-genotyping, 50% of the patients with AD, 33% of people with MCI, and 50% of cognitively healthy control subjects had at least one *APOE* ε4 allele. APOE-genotyping of the study samples was performed as described previously [24]. Participants were tested for their sense of smell for 12 odors (Sniffin’ Sticks, Heinrich Burghart GmbH, Wedel, Germany) and classified as normal, hyposmic, or anosmic.

### 2.3. Generation of OM-Derived Neurospheres and Monitoring Neurosphere Formation

OM cell lines generated from nasal biopsy were cultured in DMEM/F12 culture medium containing 10% FBS and 1% P/S until 70% confluency. Cells were seeded at 12,000 cells/cm^2^ into T25 NuncTM (Nunc, Roskilde, Denmark) flasks pretreated with poly-L-lysine (1 µg/cm^2^) (Sigma-Aldrich, St. Louis, MO, USA #P4707) and cultured in sphere-inducing medium; DMEM/F12 supplemented with 1% insulin transferrin selenium (ITS, Thermo Fisher Scientific, Waltham, MA, USA #41400-045), epidermal growth factor (EGF, 50 ng/mL, EDM Millipore, Burlington, MA, USA #GF144) and fibroblast growth factor-2 (FGF2, 25 ng/mL, EDM Millipore #GF003-AF). Growth factors were added every two days. Neurosphere formation, size, morphology and number were observed daily with an inverted light microscope (Olympus, Tokyo, Japan) and compared between the control and patient-derived cells. For neurosphere quantification, 5 phase contrast images were randomly captured under the inverted light microscope at lower magnification (×10). Neurospheres were manually counted and grouped according to their numbers < 5, 20–50 and 50–100. The size of neurospheres was determined from the same images (approximately 12–18 neurospheres per cell line) using ImageJ analysis software version 1.53k (National Institutes of Health, Bethesda, MD, USA). Typical passage number of OM cells before sphering was between 2 and 5.

### 2.4. Neurosphere Collection and Replating Cells as Olfactory Neurosphere-Derived Cultures

Neurospheres detached or loosened from the culture dish surface when they had reached their maturity at 2 to 7 days depending on the cell line. The free-floating neurospheres were harvested every second day together with a sphere-inducing culture medium, centrifuged at 100× *g* for 5 min and dissociated with TrypLE (at 37 °C for 10 min) into single cells. These dissociated ONS cells were replated at 4000–6500 cells/cm^2^ into T75 Nunc^TM^ (Nunc, Roskilde, Denmark) culture flasks and cultured in DMEM/F12 supplemented with 10% FBS until they reached 75% confluency approximately in 1–2 weeks. All the experiments here were undertaken on ONS cells at passage number below 10.

### 2.5. Immunofluorescence of Olfactory Neurosphere-Derived Cells

ONS cells were cultured on 13 mm plastic coverslips (Sarstedt, Nümbrecht, Germany) with DMEM/F12 medium containing 10% FBS and 1% P/S. Cells were fixed either in 4% paraformaldehyde (PFA) or ice-cold methanol for 15 or 5 min and washed with PBS. PFA fixed samples were permeabilised with PBS containing 0.5% Triton-X 100. Blocking was performed at RT with 2% bovine serum album (BSA) (Sigma-Aldrich, St. Louis, MO, USA #A7906), 2% goat serum (Sigma-Aldrich, St. Louis, MO, USA #G9023) and 0.1% Triton-X 100 in PBS for 2 h. Primary antibodies for DCX (1:200, Abcam, #Ab52642), TUBB3 (1:500, Biolegend, San Diego, CA, USA #801202), Nestin (1:200, Abcam, Cambridge, UK #22035), AQP4 (1:200, Abcam, Cambridge, UK #9512 and S100b (1:200, Abcam, Cambridge, UK #ab52642) were diluted in blocking solution and incubated overnight at 4 °C. The following day, cells were washed three times with 0.1% Triton-X 100 in PBS followed by incubation with secondary antibodies (Alexa Fluor −488, (#A-11029), −594 (#A-11037) and/or −647 (#A-21247), (all from Invitrogen, Waltham, MA, USA) for 2 h at RT in the dark and counterstained with a nuclear dye (Hoechst 1 µg/mL). Fluorescence images were captured using a confocal laser scanning microscope (LMS-780, Carl Zeiss AG, Oberkochen, Germany) and processed using Zeiss ZEN software version 1.1.2.0 (Carl Zeiss AG, Oberkochen, Germany).

### 2.6. Total RNA Transcriptome Analysis

Total RNA from 18 ONS cell samples (6 AD patients, 6 MCI and 6 HC) were extracted using Allprep universal kit (Qiagen cat. no. 80224). RNA-seq libraries were prepared using the Illumina TruSeq Stranded Total RNA library Prep Gold Kit (Illumina, San Diego, CA, USA 20020598) and TruSeq RNA Single Indexes (Illumina, 20020492/20020493), according to the manufacturer’s protocol (Illumina, document # 1000000040499 v00, October 2017) described briefly as follows. To enrich for mRNA, 27–100 ng of total RNA was depleted of rRNA using Ribo-Zero Gold. The enriched mRNA was then subjected to a heat fragmentation step aimed at producing fragments between 120–210 bp (average 155 bp). cDNA was synthesized from the fragmented RNA using SuperScript II Reverse Transcriptase (Invitrogen, 18064014) and random primers. The resulting cDNA was converted into dsDNA in the presence of dUTP to prevent subsequent amplification of the second strand and thus maintaining the ‘strandedness’ of the library. Following 3′ adenylation and adaptor ligation, libraries were subjected to 15 cycles of PCR to produce libraries ready for sequencing. The libraries were quantified on the Perkin Elmer LabChip GX Touch with the DNA High Sensitivity Reagent kit (Perkin Elmer, Waltham, MA, USA, CLS760672). Libraries were pooled in equimolar ratios, and the pool was quantified by qRT-PCR using the KAPA Library Quantification Kit-Illumina/Universal (KAPA Biosystems, KK4824) in combination with the Life Technologies Viia 7 real time PCR instrument. Sequencing was performed using the Illumina NextSeq500 (NextSeq control software v2.2.0/Real Time Analysis v2.4.11, Illumina, San Diego, CA, USA). The library pool was diluted and denatured according to the standard NextSeq protocol (Document # 15048776 v05), and sequenced to generate paired-end 76 bp reads using a 150 cycle NextSeq500/550 High Output reagent Kit v2.5 (Illumina, San Diego, CA, USA 20024907). After sequencing, fastq files were generated using bcl2fastq2 (v2.18.0). Library preparation and sequencing was performed at the Institute for Molecular Bioscience Sequencing Facility (University of Queensland, Brisbane, Australia).

Low quality reads and reads under 36 bp were removed with Trimmomatic, using default options. High quality reads were mapped to the Gencode release 38 transcript sequences using STAR RNA-seq aligner version 2.7.10 a, which produced binary alignment (BAM) files. The BAM files were given to Salmon version 1.6.0 in alignment-based mode to quantify the number of mapped reads per transcript in GENCODE database (v27; reference genome GRCh37), which created a transcript count data matrix for each sample. These quantification files were combined, then normalised by a read counting approach and transcripts were combined into their respective genes. Genes with less than 10 reads mapped to them across all samples were removed, then a negative binomial distribution and Fischer’s exact statistical test were used to determine differential expression (DE) using the R Bioconductor package DESeq2 between AD, MCI and HC samples. The most differentially expressed transcript was used as proxy for each gene.

### 2.7. RNA Extraction and Quantitative Real-Time Polymerase Chain Reaction (qRT-PCR)

Cells grown on tissue flasks were rinsed with PBS and exposed to TRIzol^TM^ reagent (Life Technologies, Grand Island, NY, USA). Total RNA was extracted using the Direct-zol RNA Miniprep kit (Zymo Research, Irvine, CA, USA) according to the manufacturer’s instructions. RNA quality and quantity was measured using NanoDrop^TM^ Spectrophotometer, after which RNA was converted to cDNA using SensiFAST^TM^ cDNA synthesis kit (Bioline, London, UK) according to the manufacturer’s instructions. For qRT-PCR, cDNA was diluted 1:10 to generate a working solution and mixed with SensiFAST^TM^ SYBR^®^ Lo-ROX master mix and gene-specific primers (Appendix A). The qPCR run was performed in triplicate for each sample on QuantStudio^TM^ 5 Real-Time PCR system (Thermo Fisher Scientific, Waltham, MA, USA). Ct values were normalised to Ct values of 18S endogenous control (ΔCt values), which was found to be consistent across cell lines. ΔΔCt values were calculated as 2^(−ΔCt)^ and presented as ΔΔCt multiplied by 10^6^ or as fold change.

### 2.8. Gene Enrichment and String PPI Analyses

We have applied further filters to the initial set of differentially expressed genes (defined as those with *p*-value for expression changes lower or equal than 0.01). To focus on biologically relevant genes, we selected those with a logFC either greater than 0.4 (upregulated DEGs) or lower than −0.4 (downregulated DEGs). Gene lists were retrieved for each comparison (i.e., AD vs HC, AD vs MCI and MCI vs HC) and further assessed in the functional analyses computational enviroments WebGestalt and String. We used WebGestalt 2019 as a tool to conduct over-representation analyses and find enriched gene ontologies and pathways when comparing DEG lists against the reference set. Gene ontology enrichment analyses were performed concomitantly for non-redundant biological processes and non-redundant molecular functions. Pathway enrichment analyses were performed for Reactome, Wikipathway and Panther databases. String protein–protein interaction networks were generated for each list of DEGs and enrichment was defined for functional categories with FDR lower than 0.05. Additionally, we compared DEG sets with sets of genes involved with phenotypes of interest (i.e., aging and intellectual deficiency) and assessed the statistical significance of the calculated overlaps using R (version 4.0.2) ‘newGeneOverlap’ function. All experiments were performed using the entire human genome as the statistical background, against which each list of DEG was compared. The dataset of genes related to aging has been compiled in-house combining entries from the main databases (i.e., GenAge, GenDR, LongevityMap, CellAge, AgeFactDB, JenAge, AGEMAP, OMIM) to a total of 5926 genes (available upon request).

### 2.9. Statistical Analysis

GraphPad Prism software version 8.0 (Graphpad Software Inc. San Diego, CA, USA) was used to produce graphs and statistical analysis. Comparisons between two groups were analysed using a two-tailed unpaired *t*-test with Welch’s correction. Comparisons between three groups were analysed by one-way ANOVA followed by Tukey’s post hoc tests. All data are presented as mean ± SEM and *p* ˂ 0.05 was considered significant. Statistical significance was determined as * *p* < 0.05, ** *p* < 0.01, *** *p* < 0.001, **** *p* < 0.0001, as detailed in figure legends.

## 3. Results

### 3.1. Characterization of Neurosphere Formation and Neuroglial Differentiation in AD, MCI and HC Olfactory Cultures

The formation of olfactory neurospheres was examined by culturing OM cells derived from AD, MCI and HC nasal biopsies using previously established methods [13,14]. Sphere size, morphology, number, and formation time were compared between control and patient-derived cells (Appendix A). Typical sphere formation time from day 0 to collection was ~6–7 days for disease and control cell lines (Figure 1A,B and Appendix A). There was a clear pattern of sphere formation with cells clustering at day 1, formation of networks at day 2, formation of immature and largely attached neurospheres at days 3 and 4 and typically round, consistent sized, floating, mature neurospheres formed at days 5 to 7 (Figure 1B). Sphere formation between HC, MCI and AD was largely consistent. Sphere diameter was typically between 80 μm and 150 μm (Figure 1B,C and Appendix A). Most of the mature neurospheres were round and compact and revealed a consistent size.

Patient-derived OM cells formed neurospheres similarly to those derived from HCs. There was more variation in neurosphere size and formation time within each cohort (AD, MCI and HC) than between the cohorts (Appendix A), which indicated that individual patient or biopsy-related differences likely caused changes in sphering. Two HC, one MCI and one AD line generated fewer numbers of neurospheres (<5/25 cm^2^) compared to the other lines (Appendix A), and their sphere-formation was slower. Neurosphere morphology was mainly compact and round in all cell lines, despite occasional large and abnormal cell aggregates, in particular in HC and MCI lines. The HC and MCI cells also formed larger (sphere diameter > 150 μm) neurospheres when compared to AD lines. Analysis of neurosphere sizes revealed that 35.2% of neurospheres in HC cells and 36.3% of neurospheres in MCI cells were >150 μm, whereas the corresponding percentage for AD cells was only 10.2%. The difference in size between the large HC and AD neurospheres was statistically significant, (*p*-value of 0.0237). In addition, 14.5% of neurospheres generated by AD-patient-derived OM cells were small (<60 μm), whereas only 1.3% of MCI and 2.2% of HC OM cells generated neurospheres of 60 μm or less (Appendix A).

To confirm that neurospheres contained cells capable of expressing a neuro-glial phenotype, neurospheres were dissociated into single cells and cultured as adherent neurosphere-derived cells (ONS) [13] (Figure 1D) for a further 14 days. In keeping with previous reports [14,25], ONS cells revealed neural stem cell (nestin [26] and PAX6 [27]), immature neuron (βIII-tubulin (TUBB3 [14]), microtubule-associated protein 2 (MAP2 [28]) and doublecortin (DCX [29]) and astroglial (aquaporin-4 (AQP4 [30]) and S100B [31]) expression via immunocytochemistry and qRT-PCR (Figure 1E and Appendix A). ONS cells were predominantly of immature cell neuro-glial lineage in keeping with previous reports [12,14]. However, this immature neuro-glial phenotype differed significantly from the original OM cultures in which neuronal and glial cell markers were reported to be (largely) absent [21]. No significant marker expression differences were observed between HC, MCI and AD-derived ONS cells.

### 3.2. Transcriptomic Analysis of AD, MCI, and HC Neurosphere-Derived Cells Revealed Differential Expression of Genes Associated with Cognition and Air Pollution Exposure

We performed whole transcriptome sequencing on ONS cells to determine if there were differences in gene expression between 6 AD, 6 MCI, and 6 HC individuals. Principal Component Analysis (Figure 2A) of statistically significant genes (*p* < 0.01, Appendix A), revealed clear discrimination between HC, MCI and AD patient-derived ONS cells. For subsequent analysis, we added a further *p* ≤ 0.015, log2FC < −0.4 or log2FC > 0.4 cutoff (Appendix A) to define genes as significantly differentially expressed. As a result, we recove-red 225 DEGs when comparing AD and HC groups (with 98 downregulated and 127 upregulated in AD); 431 DEGs between AD and MCI (159 downregulated and 272 upregulated in AD) and 209 between MCI and HC (123 downregulated and 86 upregulated in MCI) (Figure 2B). Less than 5% of the DEGs were detected concomitantly when comparing AD and MCI with HC. The greatest change in gene expression occurred between MCI and AD (Figure 2B,C). This is in agreement with observations made in microarray studies, (i.e., [32]), which postulate that changes in gene expression occur in different directions when the brains of MCI and AD patients are compared with healthy individuals. These opposing differences would add up to a greater distinction between the transcriptomes of MCI and AD states than between any of the states and assigned controls.

Examining the role of the top 15 gene candidates from each comparison, a substantial proportion had previously reported roles related to neural or cognitive functions (53.3%) and/or air pollution-induced changes (37.8%) (Table 1). AD vs MCI analysis revealed the highest number of both gene groups (10 related to cognition and 9 related to air pollution). One of these genes is A-kinase anchoring protein 6 (*AKAP6*)*,* which was the highest-ranked gene in the AD vs MCI, and the AD vs HC comparisons (Table 1). *AKAP6* expression was highly downregulated in AD compared to MCI and HC (Table 1), while in MCI *AKAP6* was upregulated compared to HC (*p*-value 0.038). A further five genes had cognitive or neural roles in AD vs HC, and nine in AD vs MCI (Table 1). Two of the confirmed changes were the vitamin D receptor (*VDR*) and matrix metalloprotease 2 (*MMP2*), both of which have been associated with cognitive changes and AD in numerous studies [33,34]. Nine of the top fifteen altered genes in the AD vs MCI comparison have been reported to change in response to air pollution exposure, commonly particulate matter from vehicle exhaust fumes (Table 1) (in various cell types and tissues). Seven of the genes associated with cognitive changes were also reported to be altered in response to air pollution (Table 1). *AKAP6*, the most highly altered gene in Table 1 by *p*-value (AD vs MCI, and AD vs HC), reportedly decreases in response to particulate matter air pollution exposure in humans [35].

### 3.3. AD and MCI Olfactory Neurosphere-Derived Cells Reveal Sex, ApoE, and Olfactory Function-Associated Differences in Gene Expression

We measured RNA expression in a total of 12 DEGs (from Table 1), seven DEGs from AD vs MCI comparison and six DEGs from AD vs HC comparison (one of them being the same in both comparisons) (Figure 3). Five DEGs from AD vs MCI and eight DEGs from MCI vs HC comparison from Table 1 were not tested due to low PCR amplification. qRT-PCR validation confirmed the RNA-Seq results of nine DEGs; *AKAP6* (AD vs MCI), *VDR*, *MMP2*, *AKR1C2*, *ANGPT1*, *WISP1*, *ADAM12*, *KRT19* and *SORCS2* and showed the same trend with the remaining DEGs; *PLXNA4*, *PRLR*, *SEMA6D* and *AKAP6* (AD vs HC) (Figure 3). We then examined whether there were relationships between differentially expressed genes and sex or ApoE status. When assessing cohorts as males or females, we obtained four females and males from six of each HC cohort and AD cohort, and two females and four males from the six in the MCI cohort, respectively (Table 2). *AKAP6* expression was significantly lower in AD compared to MCI and HC in males, but not between MCI and HC (Appendix A). In females, *AKAP6* expression was also significantly lower in AD compared to MCI, and lower in HC compared to MCI, but not between AD and HC (Appendix A). Likewise, there was differential gene expression between males and females across HC, MCI, and AD samples for *VDR*, *MMP2*, *ANGPT1, WISP1, KRT19* and *SORCS2* (Appendix A). We further examined whether sex differences existed for each of the 12 genes irrespective of clinical status (8 males and 10 females, Table 1, Appendix A). *AKAP6* expression was significantly higher in males than females (Appendix A). *ADAM12* expression was also higher in males than females, while no other gene examined had altered expression in all males compared to all females (Appendix A).

Examination of *APOE* status also revealed an association between the presence or absence of the high-risk *APOE4* allele and the expression of leading DEGs. We examined the expression of the 12 genes from Table 1 across AD, MCI, and HC for individuals that lacked an allele ε4 of apolipoprotein E (*APOE2/3* or *APOE3/3*) and compared this to individuals with one or two ε4 alleles (*APOE3*/*4* or *APOE4*/*4*) (Table 2). In the *APOE2*/*3* and *APOE3*/*3* groups, there was lower *AKAP6* expression in AD compared to MCI, and HC compared to MCI, but not between AD and HC (Figure 4). This was the same for *APOE3*/*4* and *APOE4*/*4*-containing individuals, although the *p*-value of the changes between AD, MCI and HC were different to that observed in the *APOE2*/*3* and *3*/*3* groups. There were, however, *APOE4*-dependent differences in gene expression across the three clinical states (HC, MCI and AD) for *VDR, MMP2, AKR1C2 ANGPT1, WISP1*, *ADAM12, PLXNA4, PRLR* and *KRT19*. Interestingly, *MMP2* revealed substantially higher expression in AD compared to MCI and HC within the *APOE2*/*3* and *3*/*3* group, while it showed no change across HC, MCI and AD in the *APOE3*/*4* and *4*/*4* group (Figure 4). In contrast to the outcomes from the groups with different clinical status, when we examined gene expression independent of clinical status, we found that *AKAP6* was significantly elevated in *APOE3*/*4* and *4*/*4* individuals compared to *APOE2*/*3* and *3*/*3* individuals (Appendix A). *MMP2, AKR1C2, ANGPT1*, *PRLR*, *SEMA6D* and *SORCS2* also revealed differences bet-ween the two different *APOE* groups, but unlike *AKAP6*, the expression of all of them was significantly lower in *APOE3*/*4* and *4*/*4* individuals compared to *APOE2*/*3* and *3*/*3* individuals (Appendix A). These findings show that leading differentially expressed genes were associated with sex (*AKAP6* and *ADAM12)* and ApoE status (*AKAP6, MMP2, AKR1C2*, *ANGPT1, PRLR* and *SORCS2)* suggesting that one or more of these genes could be explored further for their relationship to olfactory or cognitive changes in age-related disorders.

### 3.4. DEGs Associated with Aging, Intellectual Deficit, and Alternative Splicing were Strongly Affected Both in AD and MCI Patients

To further explore how the changes in the transcriptome were affecting different biological functions, we performed multiple complementary analyses (String and WebGestalt) to find pathways in which differential gene expression was enriched. The top 10 pathways that were altered in the AD vs MCI comparison are listed in Figure 5A. We performed correlation analyses between DEGs (cutoff of *p* ≤ 0.015 and log2FC < −0.4 or >0.4) identified in each comparison and datasets of genes involved with aging and intellectual deficit. We observed that genes associated with aging were strongly affected both in MCI and AD patients. In total, 67 aging-related genes were differentially expressed between MCI and HC (*p*-value of enrichment 6.9 × 10^−19^), 76 between AD and HC (*p*-value of enrichment 8.7 × 10^−23^) and 147 between AD and MCI (*p*-value of enrichment 2.2 × 10^−43^) (Figure 5B). Although this enrichment is somewhat expected, given the age status of most AD and MCI patients and the interplay between the neurological pathways that underlie these conditions, the list of genes still revealed interesting aspects of the relationship between aging and AD, especially in olfactory-derived cells. When we investigated the list of DEGs for the presence of genes associated with intellectual deficiency, we found that 26 intellectual deficiency-related genes were differentially expressed between MCI and HC (*p*-value of enrichment 4.9 × 10^−7^), 26 between AD and HC (*p*-value of enrichment 2 × 10^−6^) and 63 between AD and MCI (*p*-value of enrichment 7.9 × 10^−18^) (Figure 5C). While classical functional intellectual (i.e., developmental) deficit is not an expected feature of AD, this connection between olfactory cells and brain-related conditions highlights the close relationship between intellectual deficit and cognitive decline as well as the potential of the olfactory cell model to act as a predictor for cognitive/neural disease changes.

Using String [68], we explored the network of proteins affected by the transcriptome reprogramming that takes place in AD and MCI compared with HC. Most strikingly, aging-related genes that are differentially expressed in AD were most frequently associated with cell differentiation (31 genes, False Discovery Rate (FDR) < 0.05), and generation/development of neurons (19 genes, FDR < 0.05). In MCI patients, the previously mentioned functional categories were even more significantly affected (cell differentiation revealed 34 genes with FDR < 0.05, and generation of neurons revealed 20 genes with FDR < 0.05. We also observed evidence of altered extracellular matrix organisation and cell adhesion, (22 genes, FDR < 0.05). A related finding was the presence of a 10 protein cluster, including matrix metalloproteinases MMP2 and MMP14, which were encoded by DEGs in MCI and have been correlated with age-related conditions through their interaction with inflammatory and anabolic cytokines [69]. When we compared AD with MCI, we found that the neural functional processes that were present in the previous two comparisons were not observed. Instead, we observed 49 genes related to cell differentiation, 28 with extracellular matrix organisation and cell adhesion, and 22 with cytoskeleton organisation, which was not detected in the previous two comparisons (FDR < 0.05).

We also used String networks to investigate functionally enriched categories in the general set of DEGs for each comparison. When AD was compared with HC individuals, the PPI enrichment *p*-value was 0.021 and only one functional enrichment was observed, namely encompassing 145 genes related to alternative splicing (FDR < 0.05). When AD was compared with MCI (PPI enrichment *p*-value 1.8 × 10^−7^), again alternative splicing was the main affected mechanism (284 genes with FDR < 0.05), but adherens and anchoring junction ontologies were also highlighted as significantly affected (22 genes and FDR < 0.05). Lastly, when MCI patients were compared with controls (PPI enrichment *p*-value of 1.3 × 10^−4^), alternative splicing was once more shown to be significantly affected (with 138 genes and FDR < 0.05), as well as extracellular matrix (ECM) organisation and degradation pathways (> 20 genes and FDR < 0.05), and nervous system proteins (>80 genes and FDR < 0.05).

We have also assessed DEGs separately given their status as upregulated or downregulated. All PPI enrichment *p*-values were lower than 0.015 and the minimum number of annotated genes was 86. Our most interesting observations were that the alternative splicing pathway described above was affected in all comparisons, and was strongly upregulated in AD when compared with HC (91 genes and FDR << 0.01), and MCI (185 genes and FDR << 0.01). No significance was found when MCI was compared with HC and genes were treated separately (downregulated-only or upregulated-only). Regarding downregulated genes in AD vs MCI comparison, the most striking result was the downregulation of many protein groups related to cell adhesion, anchoring and cytoskeleton (FDR < 0.05 for all subclasses). The main difference between HC and MCI was in brain/nervous system genes and ECM organisation pathways. Most of the ECM-related genes were downregulated in MCI. Comparing AD to MCI, the DEGs were enriched in cell adhesion processes. Especially, cell adhesion genes seem to be heavily downregulated in AD, as well as cytoskeleton-related genes (predominantly actin cytoskeleton). Comparing MCI to HC, cell adhesion genes were also downregulated in MCI, as well as ECM-related genes. Thus a loss of cellular integrity and cell-to-cell communication may play a role, even in the early stages from HC to AD. Together, these observations point to chromatin and/or transcriptional component having an important role in the development of AD as well.

## 4. Discussion

One of the earliest symptoms of AD is impaired olfaction [70]. However, the molecular basis of olfactory loss remains elusive. Found deep within the nasal cavity, olfactory stem cells provide a window into the brain. Their inherent ability to form neuroglia make these cells an important model system to investigate the early pathophysiological changes that take place in AD. These cells can be obtained from patients with relative ease [13], and ONS cells are cost-effective to maintain in large quantities. These cells also provide an opportunity for longitudinal studies with multiple sampling possible from the same patient. In the present study, we report for the first time, an extensive transcriptome profiling of AD ONS cells. Our RNA-Seq results identified altered genes associated with cognitive changes, and a preponderance of highly altered DEGs reported to respond to air pollution, a known environmental risk for AD [10]. Pathway analyses revealed that DEGs associated with aging, intellectual deficit, and alternative splicing were strongly affected both in AD and MCI patients. Additionally, cytoskeleton organisation, extracellular matrix organisation, and cell adhesion were compromised in the disease state.

All of the OM cell lines used in this study were capable of forming neurospheres using established sphere-forming conditions [13,19]. This is one of the key features of stemness [71], and confirms that OM cells include stem and progenitor cells [15,72]. HC, MCI and AD neurosphere-derived ONS cells grown as adherent cultures in a serum-supplemented medium expressed the neural stem cell marker Nestin, which is typically expressed by neuronal precursors [26,73] and a key feature of stemness [71]. A neuro-glial phenotype was confirmed by qRT-PCR and immunopositive neural (DCX, TUBB3 and MAP2) and glial (S100B and AQP4) markers. These results are consistent with previously published findings by Matigian et al. [14] who reported that neurosphere-derived cells generated from patients with Parkinson’s disease and schizophrenia were immunopositive for both Nestin and TUBB3 but not for SOX2 and proteins associated with more differentiated phenotypes such as CD45 and glial fibrillary acidic protein (GFAP). Additional studies [12,15,19,25,74] have also reported the expression of Nestin, TUBB3 and MAP2 in neurosphere-derived cells in the same serum-supplemented culture conditions that were used in this study.

In addition to similar neural cell marker expression, patient-derived ONS cells and healthy controls revealed largely analogous morphology and growth characteristics in vitro. Neurosphere average size and formation time revealed greater variation within each cohort than between cohorts as shown by variance (Appendix A). Interestingly, AD patient-derived OM cells generated lower numbers of large-sized (>150 µm) and higher numbers of small-sized (<60 µm) neurospheres compared to HC and MCI patient-derived OM cells. This suggested that sphere-forming ability may be partially dysfunctional in AD olfactory cells. Scopa et al. have recently reported a smaller average diameter of neurospheres from Tg2576 transgenic AD model mice, compared to wild type mice [75]. Alternatively, AD OM cell lines may include more progenitor cells, leading to the formation of smaller neurospheres [76,77]. Some cell lines from each cohort yielded very low numbers of neurospheres and the sphere-forming was slow. This may have resulted from small variations in the quality and/or size of the original OM biopsy sample. Alternatively, some OM biopsies may include quiescent stem cells and/or progenitor cells, which have a limited capacity for self-renewal and sphere-forming [76,77].

Little is known about the transcriptome of olfactory cells in AD patients. Lampinen et al. [21] have recently reported single-cell transcriptomic data from AD patient-derived OM cells. They observed 240 DEGs (adj. *p* < 0.05) associated with AD and involved in pathways related to inflammatory processes, RNA and protein metabolism, and signal transduction. They also identified fibroblast/stromal-like cells, globose basal cell (GBC)-like cells at different states of differentiation, and myofibroblast-like cells as common cell types present in OM cell cultures of AD patients and cognitively healthy controls [21]. Culturing of OM cells into neurospheres results in a more neuro-glial environment as described above, and this may promote the differential expression of associated neuronal or glial genes that are related to cognitive processes. This provides evidence that olfactory-derived neurosphere cultures can produce valuable insights into AD-related cognitive changes, and potentially lead to unique early biomarkers of disease progression. It is also consistent with previous studies that show important impacts of olfaction on human memory and cognitive processes [78,79]. However, there currently remains no understanding of how early olfactory impairment is associated with cognitive decline in AD.

In the current study, we performed an extensive transcriptome profiling of AD and MCI patient-derived ONS cells matched with HCs to determine how a more neural-like cell differentiation state affects gene expression. Our results revealed the highest number of DEGs in AD vs MCI compared to AD vs HC and MCI vs HC. Many of the top DEGs in each comparison (mostly in AD vs MCI) have previously reported cognition-related roles supporting the importance of the cellular environment for identifying potential disease-related gene changes in AD. It was also interesting that more significant gene expression changes were observed in AD vs MCI than in the AD vs HC comparison (Table 1). This could potentially indicate that the olfactory system expresses changes that are related to disease progression from pre-AD (MCI) to fully developed AD. Notably, this could also be associated with the important environment-brain interface role of the olfactory system. This concept is further supported by the observation that nine of the top fifteen altered genes in the AD vs MCI comparison have been reported to change in response to air pollution (Table 1). Whether this is a coincidence or reflects an as yet undetermined relationship between olfactory exposure to air pollution and subsequent progression from MCI to AD requires further exploration.

The most differentially regulated gene by *p*-value in both AD comparisons (AD vs MCI, and AD vs HC) was *AKAP6* (Table 1), which has a role in cognition, and air pollution response [35,36]. *AKAP6* (also known as *mAKAP*) is commonly localised to the nuclear envelope of neurons or sarcoplasmic reticulum where it interacts with ‘protein kinase A (PKA)’ to regulate cAMP-mediated PKA signaling [80] together with other AKAPs. This signaling pathway is critical in neuronal survival and axon growth [81] and interestingly many AKAPs have been found to play a role in various nervous system disorders [82]. *AKAP6* occurs in various brain regions and in myocytes where it is responsible for cardiac remodeling [36,83]. A potential role in cognition has been reported as *AKAP6*-rs17522122 * T polymorphism has been associated with worse baseline performance in episodic memory, working memory, vocabulary, and perceptual speed [36]. *AKAP6* also revealed a strong association with sex and ApoE status (Table 1). *AKAP6* was the only one of the top ten genes altered in more than one comparison. In addition, it was very strongly downregulated in AD being the most downregulated DEG in the whole dataset in AD vs MCI comparison and the third most highly downregulated DEG in the whole dataset in AD vs HC comparison. In MCI instead, *AKAP6* expression was upregulated (*p*-value 0.038), which was also supported by our qRT-PCR results of *AKAP6*. This could indicate its potential use as an early biomarker for AD and its possible role in AD, especially MCI to AD transition. Other leading DEGs were *ADAM12* (associated with sex), *MMP2*, *AKR1C2*, *ANGPT1*, *PRLR* and *SORCS2* (ApoE status). *ANGPT1* is associated with the progression of AD, potentially through accelerated Aβ secretion [42]. *SORCS2* is associated with increased risk of AD, declined cognitive function and altered amyloid precursor protein (*APP*) processing [84] and *MMP2* is a matrix metalloproteinase that degrades extracellular Aβ. Thus, the up-regulation of *ANGPT1, SORCS2* and *MMP2* in AD ONS cells is consistent with enhanced amyloid-associated degenerative effects. Likewise, the depletion of *AKAP6* is consistent with an impairment of some executive cognitive functions as occurs in AD patients [36].

When we examined all three comparisons, AD vs MCI, AD vs HC, and MCI vs HC, alternative splicing was the most significantly altered pathway in each comparison, particularly in AD vs MCI. It was strongly upregulated in AD, but also in MCI when compared to HC. AD associated DEGs; *CELF2*, *NSRP1* (AD vs HC) and *RAVER2* (AD vs MCI) are directly involved in alternative splicing machinery and all three genes have reported roles in brain function [85,86,87]. Alternative splicing is a post-transcriptional regulatory mechanism that results in the generation of multiple pre-messenger RNAs (mRNAs) and further proteins [88]. Alternative splicing events are typically most abundant in the brain, which displays the highest mRNA isoform complexity compared to other tissues [86]. Dysregulation of alternative splicing is associated with many age-related disorders, such as AD where alternative splicing of many AD associated genes, including the *APP* [89], presenilin 2 (*PSEN2*) [90] and microtubule-associated protein Tau (MAPT) [91], are disrupted. These findings suggest that ONS cells could be highly valuable to explore very early changes in disease-associated splicing events that lead to pathological outcomes. It would also be interesting to determine how the gene pathways are altered at different stages of disease progression, and how they compare to other neurological disorders. Other remarkably altered pathways, especially in AD, were associated with cytoskeleton organisation, cell adhesion, and extracellular matrix organisation. These pathways were downregulated in a disease state suggesting that cell morphology may be compromised in AD. All of these pathways have been associated with AD in previous studies [92,93,94].

Overall, our data demonstrate the potential of patient-derived ONS cells to provide a cell-based model for AD. Importantly, this study also revealed that ONS cells derived from nasal olfactory mucous display changes that are associated with cognition and other brain related functions. Even though the small size limits generalisation to the wider population, this is essentially a pilot study requiring repetition on a larger set of participants to increase statistical power to identify DEGs in each group. Recruitment of AD patients with different stages of disease progression would also provide more information about the pattern of gene expression in moderate and severe AD. Future studies may also further explore the role of *AKAP6,* as it was the most differentially regulated gene in both AD comparisons. Remarkable is also the observation that like the expression of *AKAP6*, the expression of many other DEGs seems to occur in a different direction in MCI compared to AD, highlighting the potential use of patient-derived ONS cells for prognostic prediction and drug discovery.

## Figures and Tables

**Figure 1 cells-11-03258-f001:**
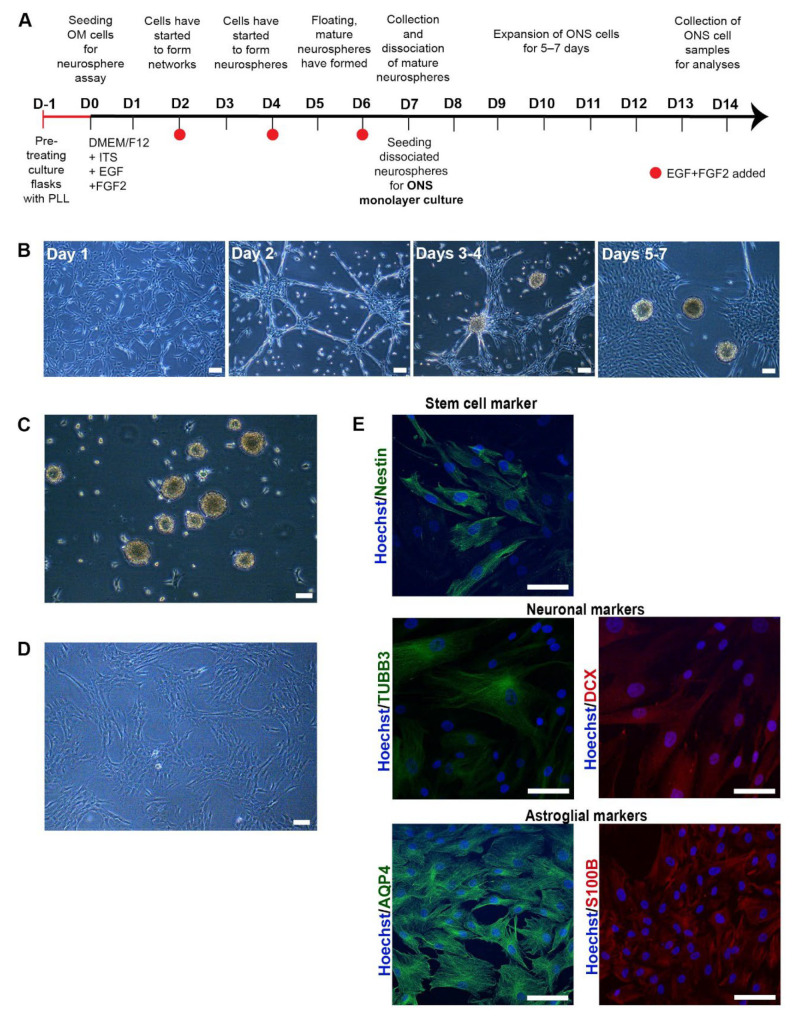
Characterisation of olfactory cells (**A**) A timeline of neurosphere formation from one day before neurosphere assay (D-1) to day 7 (D7) and expansion of ONS cells from day 7 (D7) to day 14 (D14). Culture flasks were pretreated with Poly-L-lysine (PLL) 24 h before the seeding of olfactory mucosal (OM) cells on day 0. Cells were grown in a serum-free medium containing insulin-transferring-selenium (ITS), epidermal growth factor (EGF) and fibroblast growth factor 2 (FGF2). Growth factors were added every second day until floating, mature spheres were typically ready for collection no later than day 7. Collected neurospheres were dissociated into single cells and seeded for ONS (neurosphere-derived) monolayer cell cultures that were further expanded for 5–7 days until collected for analysis. (**B**) Sphere formation from day 1 to day 7. Sphere formation took approximately 5 to 7 days in HC, MCI and AD patient-derived OM cells. Scale bar, 100 µm. (**C**) Phenotype of neurospheres. Olfactory neurospheres were formed from cells cultured from human olfactory mucosal biopsy: scale bar, 100 µm. (**D**) Olfactory neurospheres were dissociated and cultured as a monolayer where they grow in a serum-containing medium: scale bar, 100 µm. (**E**) Immunostaining of patient-derived ONS cells showed expression of characteristic markers of stem cell (Nestin (green), neuronal (βIII-tubulin (TUBB3) (green), doublecortin (DCX) (red)) and astroglial (aquaporin 4 (AQP4 (green), S100B (red)) markers. (20× Magnification, Hoechst counterstain), Scale bar, 100 µm.

**Figure 2 cells-11-03258-f002:**
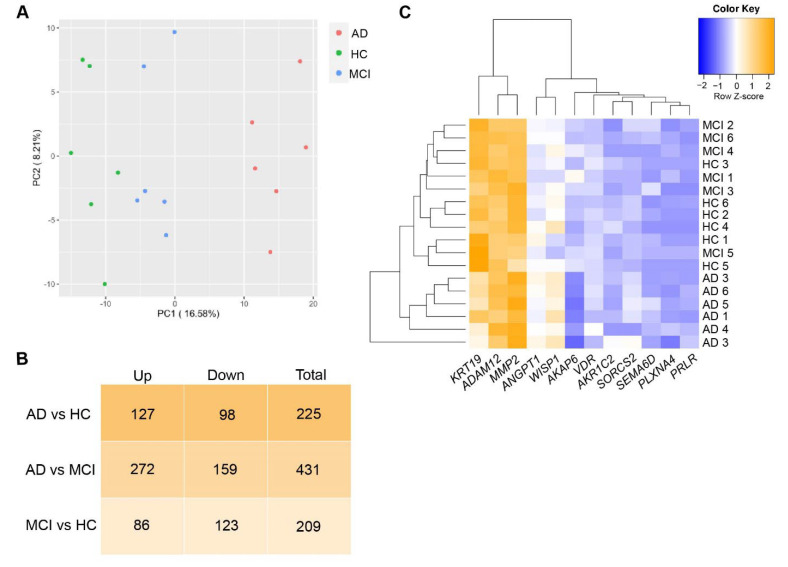
DEGs in AD, MCI and HC ONS cells. (**A**) Principal Component Analysis (PCA) plot using genes with *p* < 0.01 represent that discrimination between different cohorts: HC (green plots), MCI (blue plots) and AD (red plots) was obvious. (**B**) The number of upregulated, downregulated and total DEGs (cutoff of *p* ≤ 0.015 and log2FC < −0.4 or >0.4) in AD vs HC, AD vs MCI and MCI vs HC comparisons. The greatest change in the gene expression of all DEGs occurred between AD and MCI. (**C**) Heatmap showing Z-score normalised gene expression (log2 fold-change) of selected 12 top DEGs in ONS cells.

**Figure 3 cells-11-03258-f003:**
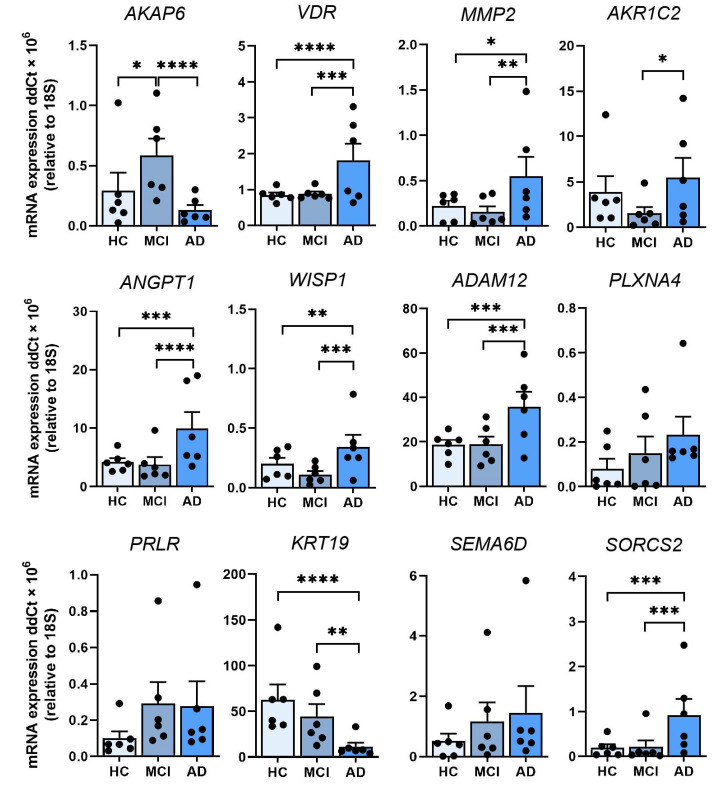
Validation of 12 leading DEGs in ONS cells by qRT-PCR. Relative expression of mRNA for *AKAP6*, *VDR, MMP2, AKR1C2, ANGPT1, WISP1, ADAM12, PLXNA4, PRLR, KRT19, SEMA6D* and *SORCS2* in HC, MCI and AD ONS cells (*n* = 6 biological replicates with 3 technical replicates per line, biological replicate data points shown) confirms the transcriptomics results. Data are presented as mean ± SEM. Statistical analysis between multiple groups was performed using one-way ANOVA, * *p* < 0.05, ** *p* < 0.01, *** *p* < 0.001, **** *p* < 0.0001.

**Figure 4 cells-11-03258-f004:**
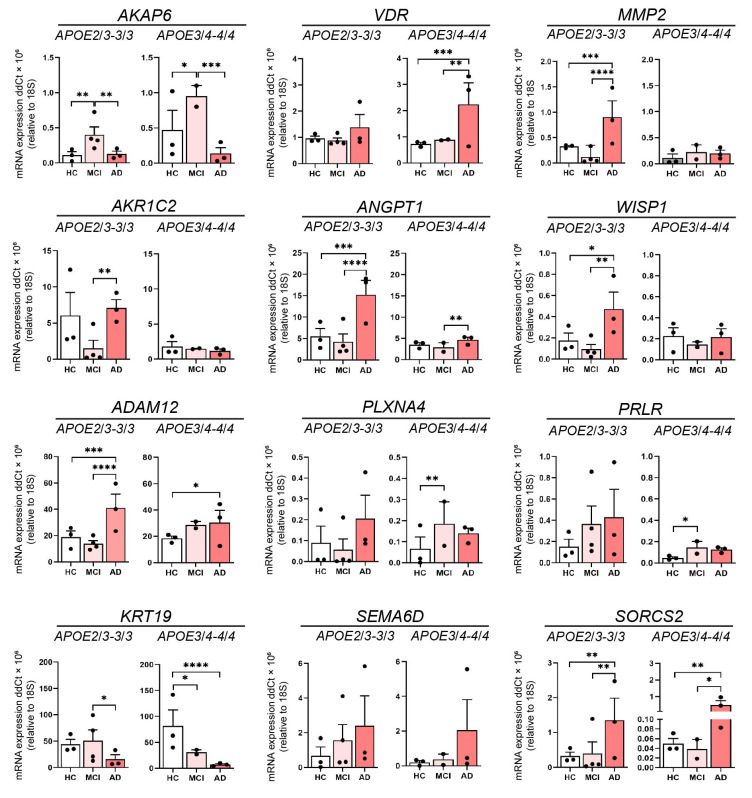
The influence of ApoE genotype of 12 leading DEGs. Relative expression of mRNA for *AKAP6, VDR, MMP2, AKR1C2, ANGPT1, WISP1, ADAM12, PLXNA4, PRLR, KRT19, SEMA6D* and *SORCS2* in HC, MCI and AD olfactory neurosphere-derived (ONS) cells (HC and AD *APOE2*/*3-3*/*3* and *APOE 3*/*4-4*/*4* genotypes: *n* = 3, MCI *APOE2*/*3-3*/*3* genotype: *n* = 4, MCI *APOE3*/*4-4*/*4* genotype: *n* = 2, 3 technical replicates per line). Data are presented as mean ± SEM. Statistical analysis between multiple groups was performed using one-way ANOVA, * *p* < 0.05, ** *p* < 0.01, *** *p* < 0.001, **** *p* < 0.0001.

**Figure 5 cells-11-03258-f005:**
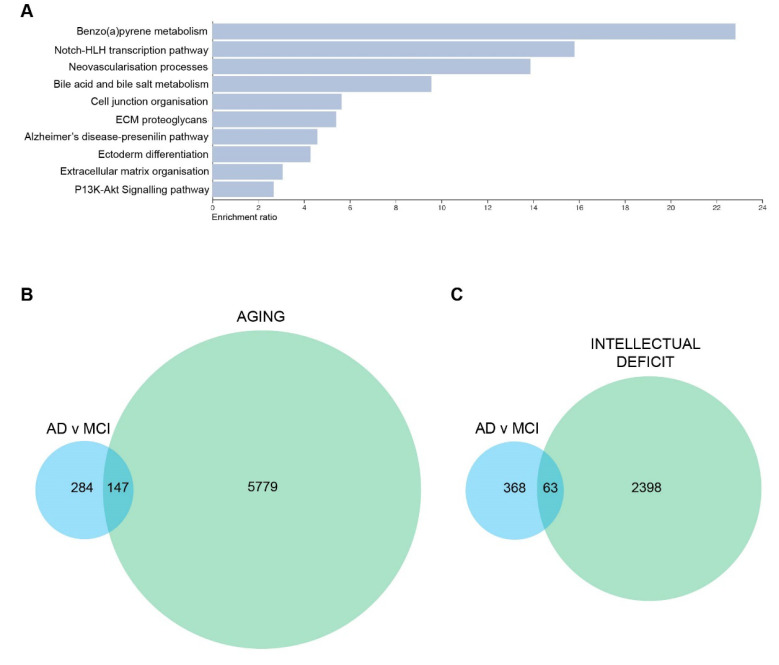
DEGs and one of the top 10 pathways in AD vs MCI comparison are associated with neuronal changes and aging. (**A**) Top 10 altered pathways in AD vs MCI comparison. (**B**) Euler diagram of DEGs associated with AD vs MCI and aging-related genes (**C**) Euler diagram of DEGs associated with AD vs MCI and intellectual deficit related genes. DEGs used to generate these figures are with a cutoff of *p* ≤ 0.015 and log2FC < −0.4 or >0.4.

**Table 1 cells-11-03258-t001:** The top significantly differentially expressed genes (DEGs) in AD compared to MCI, AD compared to HC, and MCI compared to HC as ranked by the lowest *p*-value.

Rank no. Gene	Gene Symbol (and Name)	Log2FC	*p*-Value	*p*-Adj	Change Confirmed by qRT-PCR	Differences by Sex ^2^	Differences by ApoE Genotype	Role in Cognition (Citation)	Role in Air Pollution (Citation)
**AD vs MCI**									
1	*AKAP6* (A-kinase Anchor protein 6)	−7.19	1.73 × 10^−11^	1.38 × 10^−6^	Yes	Yes	No	Yes ^3^ [36]	Yes [35]
2	*XPO6* (Exportin 6)	−0.34	9.77 × 10^−7^	0.02	ND ^1^	ND	ND	No	No
3	*MYOM1* (Myomesin 1)	−6.36	1.28 × 10^−6^	0.02	ND	ND	ND	No	Yes [37]
4	*VDR* (Vitamin D receptor)	1.39	4.19 × 10^−6^	0.04	Yes	Yes	Yes	Yes [38]	Yes [39]
5	*MMP2* (Matrix metalloproteinase 2)	1.99	5.13 × 10^−6^	0.04	Yes	Yes	Yes	Yes [34]	Yes [39]
6	*AKRIC2* (Aldo-keto reductase family 1 member C2)	5.18	5.69 × 10^−6^	0.04	Yes	No	Yes	Yes [40]	Yes [41]
7	*ANGPT1* (Angiopoietin 1)	1.83	1.27 × 10^−5^	0.06	Yes	Yes	Yes	Yes [42]	Yes [43]
8	*WISP1* (Wnt 1 inducible signaling pathway protein 1)	2.73	2.03 × 10^−5^	0.07	Yes	No	Yes	Yes [44]	Yes [45]
9	*C1QTNF1* (Complement C1q tumor necrosis factor-related protein 1	1.54	2.11 × 10^−5^	0.07	ND	ND	ND	No	Yes [46]
10	*CPXM2* (Carboxypeptidase X, M14 family member 2)	1.17	2.17 × 10^−5^	0.07	ND	ND	ND	Yes [47]	No
11	*ADAM12* (ADAM Metallopeptidase Domain 12)	0.82	2.45 × 10^−5^	0.07	Yes	No	Yes	Yes [48]	Yes [49]
14	*ZFP36L1* (ZFP36 Ring Finger Protein Like 1)	1.04	6.64 × 10^−5^	0.13	ND	ND	ND	Yes [50]	No
15	*SORCS2* (Sortilin-related VPS10 domain containing receptor 2)	3.37	8.28 × 10^−5^	0.15	Yes	Yes	Yes	Yes [51]	No
**AD vs HC**									
1	*AKAP6* (A-kinase Anchor protein 6)	−5.74	1.1 × 10^−7^	0.009	Yes	Yes	Yes	Yes [36]	Yes [35]
2	*PLXNA4* (Plexin A4)	3.34	8.68 × 10^−6^	0.2	No	No	Yes	Yes [52]	Yes [53]
3	*SHROOM2* (Shroom Family Member 2)	−2.42	2.65 × 10^−5^	0.4	ND	ND	ND	Yes [54]	Yes [55]
4	*PRLR* (Prolactin receptor)	5.25	4.49 × 10^−5^	0.6	No	No	Yes	Yes [56]	No
6	*KRT19* (Cytokeratin 19)	−2.64	0.0001	0.9	Yes	Yes	Yes	No	Yes [57]
7	*SEMA6D* (Semaphorin 6D)	5.03	0.0001	0.9	Yes	Yes	No	Yes [58]	Yes [43]
14	*SORCS2* (Sortilin-related VPS10 domain containing receptor 2)	3	0.0004	1	Yes	Yes	Yes	Yes [51]	No
**MCI vs HC**									
1	*LUPZ2* (Leusine-zipper protein 2)	4.62	2.69 × 10^−6^	0.2	ND	ND	ND	Yes [59]	No
2	*C3orf70* (Chromosome 3 Open Reading Frame 70)	2.56	3.25 × 10^−5^	0.7	ND	ND	ND	Yes [60]	No
3	*HUNK* (Hormonally Upregulated Neu-Associated Kinase)	2.71	3.27 × 10^−5^	0.7	ND	ND	ND	Yes [61]	Yes [62]
5	*ADGRL3* (Adhesion G-protein-coupled receptor L3)	5.52	0.0003	1	ND	ND	ND	Yes [63]	Yes [64]
7	*PCDH1* (Protocadherin-1)	3.2	0.0003	1	ND	ND	ND	Yes [65]	Yes [66]
8	*CPXM2* (Carboxypeptidase X, M14 family member 2)	−0.99	0.0003	1	ND	ND	ND	Yes [47]	No
11	*ID4* (Inhibitor of DNA binding 4, HLH protein)	1.80	0.0007	1	ND	ND	ND	Yes [67]	No
14	*ZFP36L1* (ZFP36 Ring Finger Protein Like 1)	−0.88	0.0008	1	ND	ND	ND	Yes [50]	No

^1^*^.^* ND = not done; ^2.^ 4 control females and 2 control males; 2 MCI females and 4 MCI males; 4 AD females and 2 AD males; ^3.^ Yes in red colour = genes that are associated with both cognition and air pollution.

**Table 2 cells-11-03258-t002:** Summary of donor information.

Study Cohorts		Healthy Control (HC)	Mild Cognitive Impairment (MCI)	Alzheimer’s Disease (AD)
No. of participants		*n* = 6	*n* = 6	*n* = 6
Sex of participants	Females (%)	66.6%	33.3%	66.6%
	Males (%)	33.3%	66.6%	33.3%
Age of participants (mean ± SD ^1^)		70 ± 5.2	72 ± 6.5	71 ± 8.3
ApoE ^2^ genotype	E2/E3 (%)	0%	16.6%	0%
	E3/E3 (%)	50%	50%	50%
	E3/E4 (%)	50%	33.3%	16.6%
	E4/E4 (%)	0%	0%	33.3%
Olfactory function	Normal (%)	50%	83.3%	16.6%
	Hyposmia (%)	50%	16.6%	50%
	Anosmia (%)	0%	0%	33.3%

^1^ SD = standard deviation; ^2^ ApoE = apolipoprotein E.

## Data Availability

RNA sequencing data are available from the European Genome-phenome Archive (EGA, https://ega-archive.org/) under the accession ID EGAS00001006594. All other data presented in the main text or the Appendix A or can be made available by contacting the corresponding author.

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
