# Peer review of "An Alzheimer’s Disease Patient-Derived Olfactory Stem Cell Model Identifies Gene Expression Changes Associated with Cognition"

_cells, 2022, doi:10.3390/cells11203258_

Round 1

Reviewer 1 Report

Review of report submitted by Laura M Rantanen et al : An Alzheimer’s disease patient derived olfactory stem cell model identifies gene expression changes associated with cognition.

Authors arguments suggest that detected patterns of changes of gene expression are associated with cognitive decline in MCI and in CDR1 AD. Further studies may provide information about pattern of gene expression in moderate and severe AD. This is not a critique of a conceptually and technically novel study but a suggestion of results discussion in a broader sense and suggestion of necessity of similar studies in different stages of progression of neurodegeneration and progressive loss in AD, versus other neurodegenerative diseases with own patterns of clinical, including cognitive decline.

Cognitive decline versus intellectual deficit. The list of Keywords includes cognitive impairment as well as intellectual deficiency.

In studies of developmental abnormalities, the deficit reflects developmental shortage (partial absence) of function (functional deficit, intellectual deficit), or structural shortage or deficit (for example: developmental shortage of neurons or developmental neuronal deficit).

Neurodegenerative disorders result in a process of intellectual (functional decline) and in a process of structural decline (decrease of the number of neurons contributing to functional/cognitive decline).

The reviewer, and probably the reader will have a problem with considering AD functional and structural decline as deficit.

Example illustrating mechanistic and terminology concerns. What terminology will authors use in similar study of trisomy 21/Down syndrome cohort? These children and less than 30-year-old adults are diagnosed with developmental intellectual deficit reflected in IQ 10 to 70, as well as 28% of developmental neuronal deficit. However, at age of 40-70 years all these individuals develop Alzheimer disease and are diagnosed with AD-associated cognitive decline leading to dementia and structural changes with neuronal loss, resulting in decline of the number of neurons in almost the same magnitude as developmental neuronal deficit.

Again, this is not critique of a great report. This reflects potential collision between different techniques/terminology and interpretation. Clarification of using word intellectual deficits for AD individuals is suggested (cognitive decline is commonly accepted).

Suggested minor corrections.

Line 36. To eliminate repetitions please replace …early AD-associated disease pathways with “changes of early AD-associated pathways.”

Line 99. Voluntary study patients … suggested replacement with: Volunteers clinically diagnosed with MCI (n=6) and AD mild dementia (CDR 1) (n=6) were recruited together with cognitively healthy control subjects (n=6). Their biopsy samples were collected and processed as previously described ((28).

Line 615. Suggested replacement of the “ONS cells from the human nose” … with more anatomically/histologically informative “ONS cells derived from nasal olfactory mucous”

Reviewer 2 Report

 More details on how patients and controls were diagnosed and chosen should be added, because sampling patients is critical for the value of these studies.

Why is there a difference in genetic expression between MCI and AD?

The variability within the same group of nanospheres is high, probably because the sampling is low, or due to an intrinsic variability of the patients.

In statistics, introduce the standard variation instead of the standard error.
